

# Transcriptome analysis of transcription factors and enzymes involved in monoterpenoid biosynthesis in different chemotypes of *Mentha haplocalyx* Briq

Xin An[1], Jingqiong Wan[2], Hui Jiang[2], Yangzhen Liao[1], Chang Liu[3], Yuan Wei[2], Chongwei Wen[2] and Zhen Ouyang[1,2]

[1] School of Food and Biological Engineering, Jiangsu University, Zhenjiang, China
[2] School of Pharmacy, Jiangsu University, Zhenjiang, China
[3] Nanjing Institute for Comprehensive Utilization of Wild Plants, Nanjing, China

## ABSTRACT

**Background:** The main active ingredients of *Mentha haplocalyx* Briq. essential oils are monoterpenes. According to the component of essential oils, *M. haplocalyx* can be divided into different chemotypes. Chemotype variation is widespread in *Mentha* plants but its formation mechanism is unclear.

**Methods:** We selected the stable chemotype *l*-menthol, pulegone, and carvone of *M. haplocalyx* for transcriptome sequencing. To further investigate the variation of chemotypes, we analyzed the correlation between differential transcription factors (TFs) and key enzymes.

**Results:** Fourteen unigenes related to monoterpenoid biosynthesis were identified, among which (+)-pulegone reductase (PR) and (−)-menthol dehydrogenase (MD) were significantly upregulated in *l*-menthol chemotype and (−)-limonene 6-hydroxylase was significantly upregulated in carvone chemotype. In addition, 2,599 TFs from 66 families were identified from transcriptome data and the differential TFs included 113 TFs from 34 families. The families of bHLH, bZIP, AP2/ERF, MYB, and WRKY were highly correlated with the key enzymes PR, MD, and (−)-limonene 3-hydroxylase (L3OH) in different *M. haplocalyx* chemotypes ($r > 0.85$). The results indicate that these TFs regulate the variation of different chemotypes by regulating the expression patterns of PR, MD, and L3OH. The results of this study provide a basis for revealing the molecular mechanism of the formation of different chemotypes and offer strategies for effective breeding and metabolic engineering of different chemotypes in *M. haplocalyx*.

# INTRODUCTION

The genus *Mentha* is a perennial herb of the Lamiaceae family. *Mentha*, also known as mint, is widely cultivated for its essential oil often used in food, pharmaceuticals, condiments, and cosmetics. *Mentha* is mainly distributed in the temperate regions of the northern hemisphere, and a few are found in the southern hemisphere (*Anwar et al., 2019*; *Harley & Brighton, 1977*; *Mkaddem, Boussaid & Fadhel, 2007*). Chemotype is a form of

Corresponding author
Zhen Ouyang,
zhenouyang@ujs.edu.cn

biodiversity within essential oil-producing plant species like *Mentha* based on their chemical profiles (*Guo et al., 2008*; *Hua et al., 2009*). *Harley & Reynolds (1992)* reported nine chemotypes of five *Mentha* species: *Mentha arvensis, Mentha aquatic, Mentha longifolia, Mentha spicata*, and *Mentha suaveolens* (*Soilhi et al., 2019*). *Kofidis, Bosabalidis & Kokkini (2004)* found four different chemotypes in *Mentha Spicata* from Greece: linalool, carvone/dihydrocarvone, piperitone oxide/piperitenone oxide, and menthone/ isomenthone/pulegone. In China, *Chou, Zhou & Zhou (1995)* divided *Mentha sachalinensis* into two chemotypes: piperitone oxide/piperitenone oxide and piperitone oxide/piperitenone oxide/pulegone/menthone. Because of their chemodiversity, the occurrence of chemotypes is widespread in aromatic species and there can be different blends of essential oil components within a species. Chemotype variation is particularly common in *Mentha* (*Harley & Reynolds, 1992*; *Kofidis, Bosabalidis & Kokkini, 2004*; *Soilhi et al., 2019*).

*Mentha haplocalyx* Briq. is a widely distributed species found in Eurasia, Australia, South Africa, and China (*Dong, Ni & Kokot, 2015*). Studies on *M. haplocalyx* have mainly focused on the determination of ingredients and its biological activity. Monoterpenoids are the main constituents of the essential oil of mint. The composition and content of monoterpenes differ in different chemotypes, as do their biological activities (*Kowalczyk et al., 2021*). As one of the representative monoterpenoids of *l*-menthol chemotype mint, (−)-menthol has analgesic, antitussive, bacteriostatic, anticancer, anesthetic, osmotic, chemopreventive, and immunomodulatory effects (*Kamatou et al., 2013*). Global menthol production is approximately 34,000 metric tons per year, with the share of synthetic menthol being approximately 60% of this. However, natural menthol is generally preferred because the scent of synthetic *l*-menthol is influenced by contaminants that arise during the crystallization process (*Dylong et al., 2022*; *Lange et al., 2011*). In addition, improving the composition of the oil has focused on reducing the accumulation of (+)-menthofuran and its intermediate product (+)-pulegone (the main component of pulegone chemotype mint) (*Grulova et al., 2015*).

The essential oil of *M. haplocalyx* consists mostly of (−)-menthol, (−)-menthone, pulegone, carvone, limonene, and other monoterpenoids and its biosynthetic pathway and related biosynthetic enzymes have been identified. Monoterpenoid biosynthesis originates from the upstream methylerythritol 4-phosphate (MEP) pathway and is catalyzed by a series of enzymes (Fig. 1). In the genus *Mentha*, the precursor geranyl diphosphate is cyclized to (−)-limonene by (−)-limonene synthase (LS). In *M. Spicata*, the main component of which is (−)-carvone, (−)-limonene is hydroxylated at the C6 position and oxidized by (−)-trans-carveol dehydrogenase (CD) to produce carvone. However, in *Mentha×piperita*, the main components of which are (−)-menthone and (−)-menthol, and *M. aquatica*, the main component of which is (+)-menthofuran, the most common oxidation reaction occurs at C3. Isopiperitenol dehydrogenase (iSPD) further oxidizes (−)-trans-isopiperitenol to the first oxygen-containing intermediate (−)-isopiperitenone. Next, under the continuous reduction and isomerization reaction of (−)-isopiperitenone reductase (iSPR) and (+)-*cis*-isopulegone isomerase (iSPI), the intermediate (+)-pulegone is produced. The main reaction in *M. aquatica* is the cyclization of (+)-pulegone into

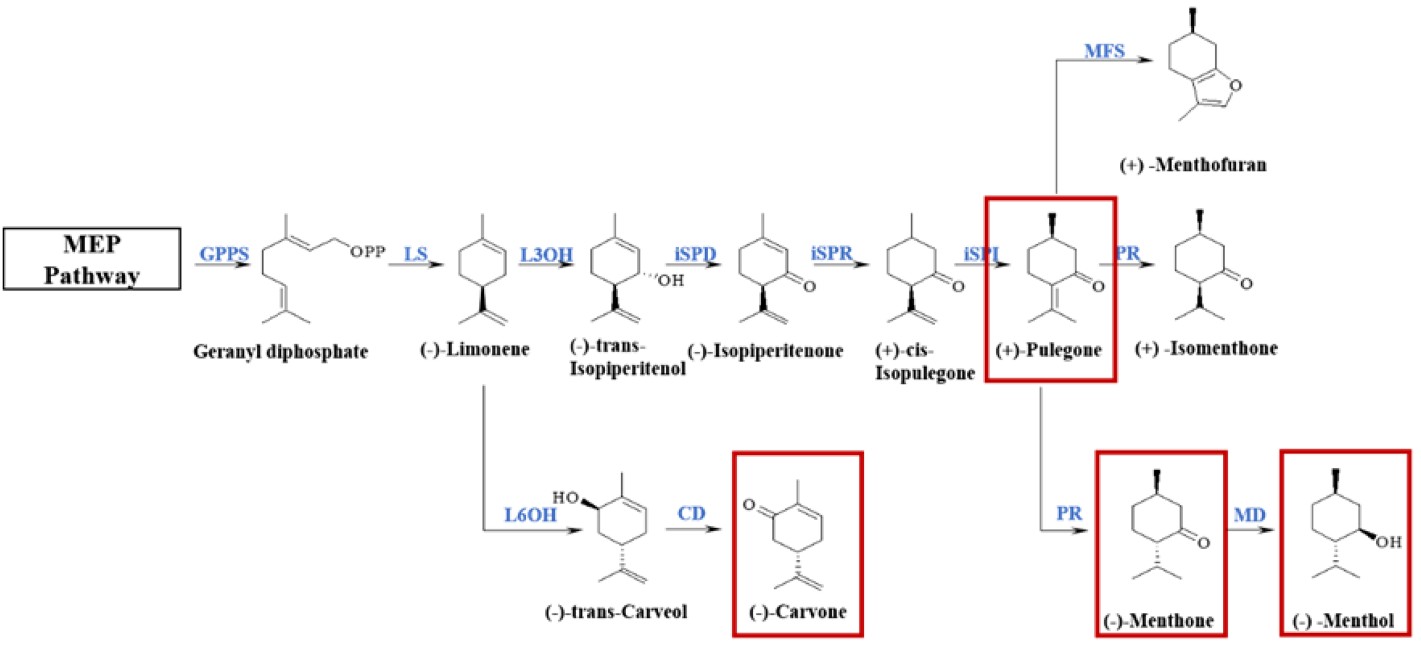

**Figure 1 Biosynthetic pathways of main monoterpenoids in genus *Mentha*.** The red boxes circle the main components of essential oils of different chemotypes classified in the early stage. The enzymes involved in this pathway are geranyl-diphosphate synthase (GPPS); (−)-limonene synthase (LS); (−)-limonene 6-hydroxylase (L6OH); (−)-trans-carveol dehydrogenase (CD); (−)-limonene 3-hydroxylase (L3OH); isopiperitenol dehydrogenase (iSPD); (−)-isopiperitenone reductase (ISPR); (+)-*cis*-isopulegone isomerase (iSPI); (+)-menthofuran synthase (MFS); (+)-pulegone reductase (PR); (−)-menthol dehydrogenase (MD).

(+)-menthofuran by (+)-menthofuran synthase (MFS), and, at the same time, a series of reduction reactions are produced by (+)-pulegone reductase (PR) and (−)-menthol dehydrogenase (MD). In *M. arvensis*, the main component of which is (−)-menthol, menthone is finally converted to menthol in the presence of MD (*Ahkami et al., 2015*; *Akhtar et al., 2017*). The composition of essential oil is regulated at multiple levels, including the abundance of transcripts, the catalytic properties of enzymes, and the epigenetic processes of specific cell types (*Ahkami et al., 2015*). However, the specific molecular mechanism of monoterpenoid synthesis in different chemotypes remains unclear.

The volatile terpenoid biosynthesis is mostly regulated at the transcriptional and post-transcriptional levels (*Tholl, 2015*). The regulation of plant terpenoid biosynthesis usually divided into two categories during development, the temporal and spatial regulation, in response to biotic and abiotic factors such as light intensity, insect/pathogen damage, nutrients, humidity and temperature conditions (*Hakola et al., 2006*; *Van Poecke, Posthumus & Dicke, 2001*). Multiple studies have shown that increased terpenoid production under different abiotic stresses is often mediated by increased transcriptional activity of specific terpenoid biosynthetic genes (*Abbas et al., 2017*). Transcription factors (TFs) regulate the coordinated expression of enzyme genes involved in the biosynthetic pathways of secondary metabolism. TFs have important functions in the synthesis of various components of medicinal plants, such as terpenoids, flavonoids, and alkaloids (*Feller et al., 2011*; *Khosla et al., 2014*; *Shi et al., 2019*). The TF families WRKY, ARF, MYB,

and bHLH may be involved in the regulation of some processes in *Mentha* plants such as *M. arvensis, M. spicata*, and *Mentha×piperita* secondary metabolite biosynthesis and plant stress response; however, regulatory mechanisms have not been defined (*Akhtar et al., 2017*; *Reddy et al., 2017*; *Wang et al., 2016*).

According to relative percent of the main characteristic components in the essential oils, 57 batches of *M. haplocalyx* samples from different origins were divided into four chemotypes in our previous investigation, including *l*-menthol chemotype ((−)-menthol content up to 78.96%), pulegone chemotype ((+)-pulegone content up to 51.77%), carvone chemotype ((−)-carvone content up to 52.25%), and *l*-menthone chemotype ((−)-menthone content up to 75.63%) (*Ye, 2015*). In a recent study, we found that *l*-menthol chemotype, pulegone chemotype, and carvone chemotype remained stable during the traditional harvest period (July to October) (*Yang, 2020*). The aim of this study was to explore the role of TFs in determining different chemotypes and to provide a basis for cultivating more widely used chemotypes of *M. haplocalyx*. The chemotypes in the stable period were selected for transcriptome sequencing analysis. Preliminary screening of related enzymes and TFs in the monoterpenoid biosynthesis pathway was performed based on the comparative transcriptome sequencing data. Correlation network analysis was performed to reveal the expression patterns of key enzyme genes and related TFs in the monoterpenoid biosynthesis pathway of different chemotypes of *M. haplocalyx*.

## MATERIALS AND METHODS

### Plant materials

*M. haplocalyx* samples (*l*-menthol chemotype, pulegone chemotype, and carvone chemotype) from the Medicinal Botanical Garden of Jiangsu University, in Zhenjiang City, Jiangsu Province, with geographic coordinates of N32°12′5″, E119°30′3″, were collected approximately every 10 days from May 10, 2020, from 12:00 to 14:00 for 5 months. Each sample had three biological replicates. The above-ground parts were dried at room temperature for the determination of essential oil content. The top leaves were collected, immediately wrapped in aluminum foil and frozen in liquid nitrogen, and then stored at −80 °C for transcriptome sequencing and chemotype variation research. Table S1 shows the samples and local temperature.

### Determination of characteristic terpenoids in essential oils

The preparation of essential oil followed the method outlined in Chinese Pharmacopoeia (2020 edition) (*Chinese Pharmacopoeia Commission, 2020*). The contents of the essential oils were determined using the method used by our group in a previous study using gas chromatography (GC) (*Yang, 2020*; *Yang et al., 2021*). GC conditions were as follows: the DB-1 column was 60 m × 0.25 mm × 0.25 μm in size and the inlet and detector temperature were both 250 °C. The split ratio was 40:1, the carrier gas was $N_2$ (99.999%), and the carrier gas flow rate was 1 mL/min. The heating program for *l*-menthol chemotype and pulegone chemotype was as follows: the initial temperature was 50 °C for 28 min, increased to 90 °C at 10 °C/min, increased again to 150 °C at 3 °C/min, increased to the final 230 °C at 10 °C/min, and held for 5 min. The heating program for carvone chemotype

was as follows: the initial temperature was 60 °C for 4 min, increased to 80 °C at 2 °C/min, increased to 150 °C at 3 °C/min, then increased to 230 °C at 5 °C/min, and held for 5 min. All determinations were performed in triplicate.

## cDNA library construction and transcriptome sequencing

The stable *l*-menthol chemotype, pulegone chemotype, and carvone chemotype samples collected on August 5[th], 2020 and pulegone chemotype sample collected on May 20[th], 2020 were selected for transcriptome sequencing. There were three biological replicates for each type. Independent total RNA extracts were prepared from the 12 samples and equal amounts of RNA from each sample were used to generate RNA-Seq libraries. The total RNA of *M. haplocalyx* chemotypes was extracted using Trizol reagent (Sangon Biotech, Shanghai, China). The quality was assessed using 1% agarose gel electrophoresis and a Qubit 2.0 RNA kit (Life Technologies, Carlsad, CA, USA). Table S2 shows the RIN value. A total amount of 2 μg RNA per sample was used as input material for the RNA sample preparations. Sequencing libraries were generated using Hieff NGS™ MaxUp Dual-mode mRNA Library Prep Kit for Illumina® (Yeasen Biotechnology, Shanghai, China) following manufacturer's recommendations. First strand cDNA was synthesized using random hexamer primer and M-MuLV Reverse Transcriptase (RNase H-) (Yeasen Biotechnology, Shanghai, China). Second-strand cDNA synthesis was subsequently performed using DNA polymerase and RNase H. Remaining overhangs were converted into blunt ends *via* exonuclease/polymerase activities. After adenylation of 3′ ends of DNA fragments, adaptor was ligated to prepare for library generation. In order to select cDNA fragments of a preferred 150–200 bp length, the library fragments were purified using the AMPure XP system (Beckman Coulter, Beverly, MA, USA). Then, 3 μL USER Enzyme (NEB, USA) was used with size-selected, adaptor-ligated cDNA at 37 °C for 15 min followed by 5 min at 95 °C before PCR. PCR was performed using Phusion High-Fidelity DNA polymerase, Universal PCR primers, and Index (X) Primer. Finally, PCR products were purified (AMPure XP system) and library quality was assessed on the Agilent Bioanalyzer 2100 system. The libraries were then quantified and pooled. Paired-end sequencing of the library was performed on the Illumina Novaseq 6000 (Illumina, San Diego, CA, USA).

## Transcriptome assembly and annotation

The original reads obtained by RNA sequencing were filtered with Trimomatic (version 0.36) to remove low-quality reads and the remaining clean reads were reassembled into transcripts by Trinity (version 2.0.6) with default settings to obtain unigenes for further analysis. Transcripts with a minimum length of 200 bp were clustered to minimize redundancy. All data generated were saved in the National Center for Biotechnology Information (NCBI) and can be accessed in the Short Read Archive (SRA) sequence database (accession number PRJNA795820). An online implementation of RNASeqPower is available at https://rodrigo-arcoverde.shinyapps.io/rnaseq_power_calc/. Unigenes were blasted against NCBI Nr (NCBI non-redundant protein database), SwissProt (A manually annotated and reviewed protein sequence database), TrEMBL (A supplement to the

SwissProt database), CDD (Conserved Domain Database), PFAM (Protein family), and KOG (Eukaryotic Orthologous Groups) databases (E-value < 1e−5).

## Analysis of differentially expressed genes

In RNA-seq analysis, we introduced Transcripts Per Million (TPM) to measure the proportion of a certain transcript in the RNA pool. We used Salmon (version 0.8.2) to calculate unigene readings and TPM. DESeq (1.26.0) was used to analyze differential expression. In order to obtain the significant differential gene, the screening conditions were set as follows: Q value < 0.05 and difference multiple |FoldChange|>2. We have chosen the groups M8 *vs* P8, M8 *vs* C8, and M8 *vs* P8 for analysis.

## Quantitative real-time PCR

The qRT-PCR method was used to verify the expression of selected genes in *M. haplocalyx*. The BeyoRT$^{TM}$ II kit was used for first-strand cDNA synthesis (Beyotime Biotechnology, Shanghai, China) from total RNA. AceQ Universal SYBR qPCR Master Mix (Vazyme Biotech Co., Nanjing, China) was used to perform the qRT-PCR reaction. qRT-PCR was conducted using the Light Cycler 96 Real-Time PCR system (Roche, Basel, Switzerland) with a 20 μL reaction mixture. The mixture consisted of 10 μL 2 × AceQ Universal SYBR qPCR Master Mix, 2 μL primer mix, 2 μL diluted cDNA, and 6 μL ddH2O. The qRT-PCR program comprised a preincubation step at 95 °C for 600 s followed by PCR: 40 cycles at 95 °C for 10 s and 60 °C for 30 s followed by a melting cycle: 95 °C for 15 s, 60 °C for 60 s and 95 °C for 15 s. Primer 3.0 online software (http://primer3.ut.ee/) was used to design primers. Expression levels of differentially expressed genes (DEGs) were determined by the method of $2^{-\Delta\Delta Ct}$ (*Liu et al., 2020*). We selected the *β*-actin gene as the reference gene (*Qi et al., 2018*). All results are representative of three independent experiments. Table S3 lists the qRT-PCR primers.

## Data analysis

The data are expressed as mean ± standard deviation (SD) and differences were analyzed by one-way ANOVA. The visualization of correlation analysis data was performed using Cytoscape (version 3.9.1) software.

# RESULTS

## Dynamic changes of characteristic terpenoids in essential oil of *M. haplocalyx* chemotypes

As shown in Fig. 2, the content of characteristic monoterpene components of different chemotypes was low and unstable until July. With the growth of *M. haplocalyx*, the chemotypes showed a stable trend and *l*-menthol, pulegone, and carvone content levels were all high in different chemotypes in August. However, in May, the menthone content was higher in pulegone chemotype. During the growing period, (−)-menthol was the main component in *l*-menthol chemotype, accounting for 44.20–51.71% of the total essential oil (Fig. 2A). During the harvest period (July to October), pulegone was the main component in pulegone chemotype, accounting for 32.41–54.10% of the total essential oil (Fig. 2B).

Peer

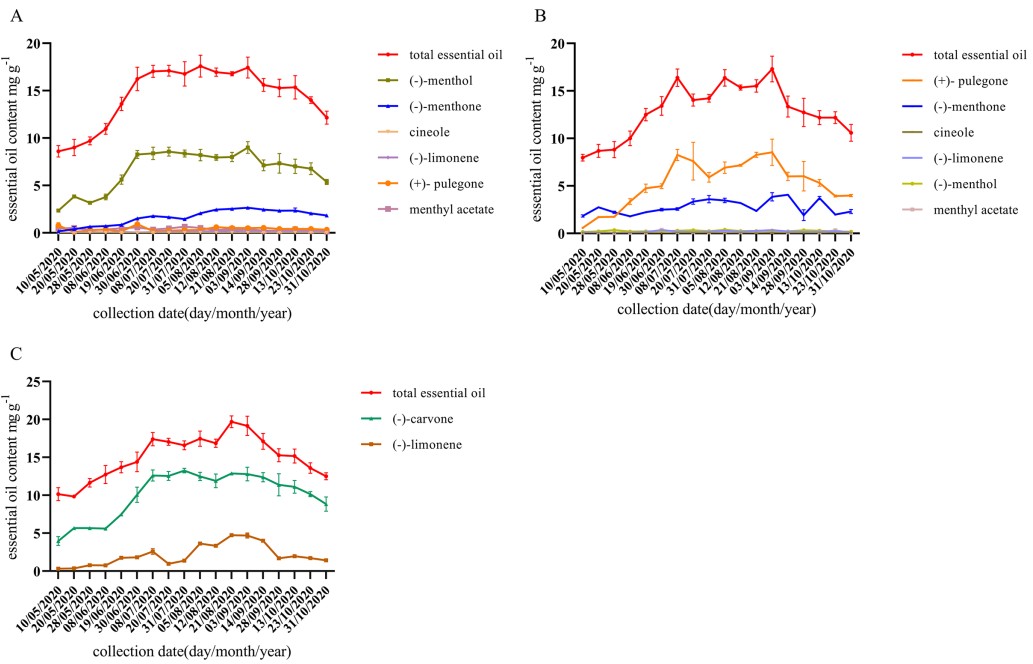

**Figure 2 Dynamic changes of essential oil components during different growth stages of different chemotype.** (A) *l*-menthol chemo type. (B) Pulegone chemotype. (C) Carvone chemotype. Datas are presented as means ± SD (*n* = 3).

Carvone was the main component in carvone chemotype during the growing period, accounting for 65.53–79.93% of the total essential oil (Fig. 2C).

## Transcriptome sequencing and analysis

We selected stable *l*-menthol chemotype (M8), pulegone chemotype (P8), and carvone chemotype (C8) samples collected in August 2020 and pulegone chemotype (P5) samples collected in May 2020 to construct 12 cDNA libraries. Transcriptome sequencing depth was 6G. The result of RNA-Seq Power analysis was 0.84. In the transcriptome data, the Q20 and Q30 values of each sample were greater than 98.41% and 94.53%, respectively. The GC content of the 12 samples was 49.14–55.01%. In addition, the mapped reading of each sample was not less than 92.48%, indicating high-quality RNA-seq data (Table 1).

In total, 254,942 unigenes were generated. The unigenes obtained showed a total length of 128,420,176, a mean length of 503.72 bp, and a mean N50 of 621 bp (Table S4). There were 103,991 unigenes annotated in the Nr database, accounting for 40.79% of total genes; 88,978 (34.90%) unigenes annotated in the SwissProt database; 52,541 (20.61%) unigenes annotated in the KOG database; and 9,252 (3.63%) unigenes annotated in the KEGG database. In total, there were 121,796 (47.77%) unigenes that could be annotated in at least one database (Table 2).

## Differentially expressed genes and enrichment analysis

From the statistical results, in the comparisons of M8 *vs* P8, M8 *vs* P5, and M8 *vs* C8, 9,676 (5,747 upregulated, 3,929 downregulated), 11,551 (7,663 upregulated, 3,888 downregulated), and 3,432 (2,026 upregulated, 1,406 downregulated) DEGs were obtained,

**Table 1  Summary of RNA-seq data.**

| Samples | Raw reads | Clean reads | Q10 | Q20 | Q30 | GC bases ratio (%) | Reads maped |
|---|---|---|---|---|---|---|---|
| M8-1 | 60555040 | 57403386 | 100.00% | 98.62% | 95.03% | 49.28% | 53437186 (93.09%) |
| M8-2 | 56240612 | 52556832 | 100.00% | 98.54% | 94.84% | 49.14% | 48875261 (93.00%) |
| M8-3 | 54735610 | 49868730 | 100.00% | 98.49% | 94.69% | 49.96% | 46493922 (93.23%) |
| P8-1 | 56426496 | 49427502 | 100.00% | 98.53% | 94.85% | 52.29% | 46239727 (93.55%) |
| P8-2 | 55271790 | 29272020 | 100.00% | 98.52% | 94.79% | 53.92% | 27070386 (92.48%) |
| P8-3 | 63262822 | 43175564 | 100.00% | 98.58% | 94.97% | 54.02% | 40342738 (93.44%) |
| C8-1 | 59170590 | 53965026 | 100.00% | 98.57% | 94.94% | 51.76% | 50418526 (93.43%) |
| C8-2 | 72506712 | 42578750 | 100.00% | 98.47% | 94.69% | 55.01% | 39654556 (93.13%) |
| C8-3 | 59742224 | 54688098 | 100.00% | 98.56% | 94.87% | 50.78% | 50942761 (93.15%) |
| P5-1 | 60516572 | 53354066 | 100.00% | 98.48% | 94.67% | 51.78% | 49859597 (93.45%) |
| P5-2 | 55042390 | 49373084 | 100.00% | 98.41% | 94.53% | 52.18% | 46293551 (93.76%) |
| P5-3 | 56660214 | 44364928 | 100.00% | 98.43% | 94.60% | 54.39% | 41583427 (93.73%) |

**Table 2  Functional annotation of unigenes.**

| Database | Number of genes | Percentage (%) |
|---|---|---|
| Annotated in CDD | 66979 | 26.27 |
| Annotated in KOG | 52541 | 20.61 |
| Annotated in Nr | 103991 | 40.79 |
| Annotated in NT | 49827 | 19.54 |
| Annotated in PFAM | 44196 | 17.34 |
| Annotated in SwissProt | 88978 | 34.90 |
| Annotated in TrEMBL | 101624 | 39.86 |
| Annotated in GO | 100452 | 39.4 |
| Annotated in KEGG | 9252 | 3.63 |
| Annotated in at least one database | 121796 | 47.77 |
| Annotated in all database | 3796 | 1.49 |
| Total genes | 254942 | 100 |

respectively (Fig. S1). The numbers of upregulated and downregulated genes in the three groups (M8 *vs* P8, M8 *vs* P5, M8 *vs* C8) were 1,139 and 670, respectively (Fig. 3).

Through GO enrichment analysis to identify the biological functions of up- and down-regulated DEGs in different comparison combinations, we found that the significant enrichment of upregulated and downregulated DEGs was divided into three categories: biological processes, cell composition, and molecular function (Fig. 4; Table S5). The top 30 pathways with the highest degree of enrichment from the KEGG enrichment analysis are shown in Fig. 5 (Table S6). Terpenoid backbone biosynthesis was significantly enriched in the three comparison groups; thus, this pathway may have a crucial function in the variation of different chemotypes. For the DEG intergroup comparison scheme used in Figs. 4 and 5, we chose the groups M8 *vs* P8, M8 *vs* C8, M8 *vs* P8.

Peer**J**

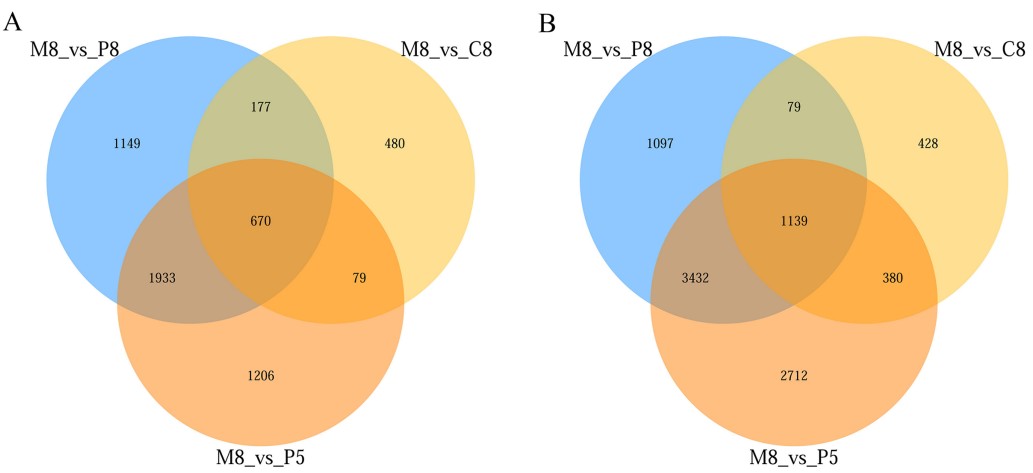

**Figure 3 Venn diagram of the three-group comparison of M8 *vs* P8, M8 *vs* P5, and M8 *vs* C8.** (A) Down-related and (B) up-related genes.

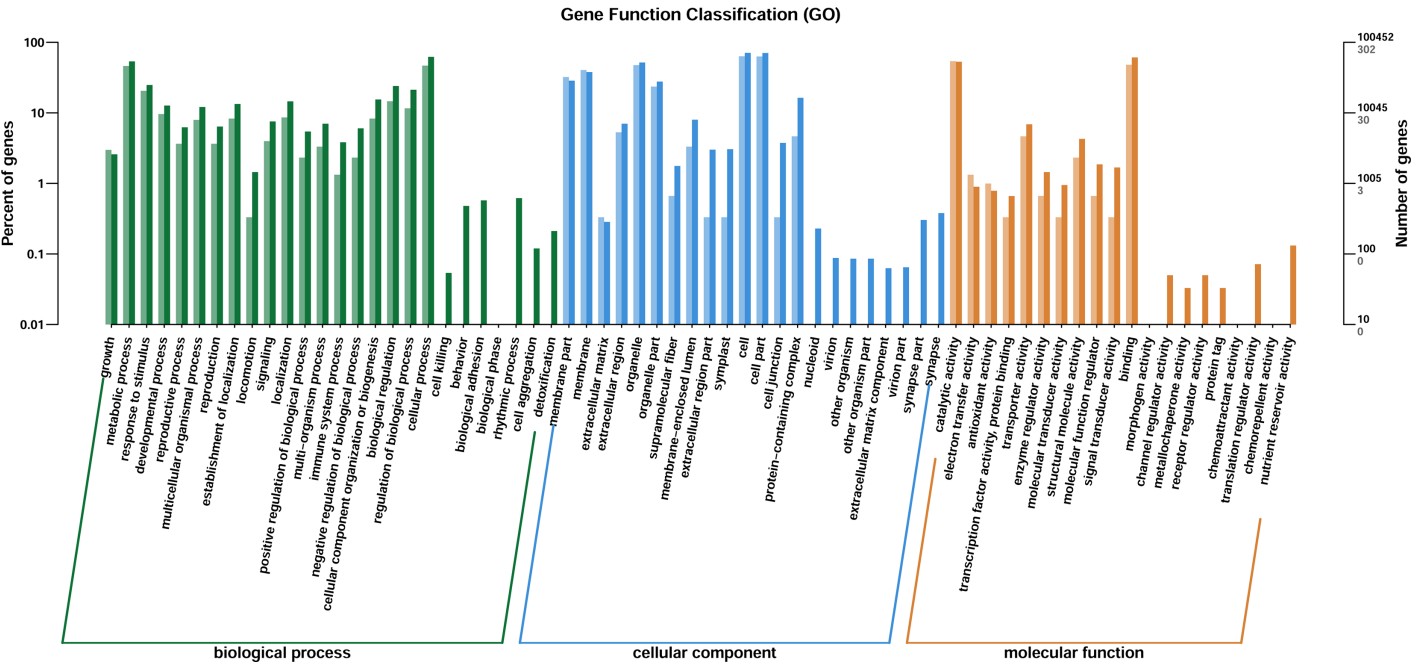

**Figure 4 DEGs of GO annotation classification.** The horizontal axis is classified by function, the longitudinal axis is the number of genes in the classification (right) and the percentage of the total number of genes on the annotation (left). Different colors represent different classifications. Light colors represent differential genes, dark colors represent all genes.

## Identification and analysis of DEGs in monoterpenoid biosynthetic pathways of different chemotypes

Ten unigenes were selected for qRT-PCR to verify the reliability of RNA-seq data. Fig. S2 shows that the qRT-PCR expression patterns of most selected unigenes closely matched the TPM results of RNA-seq data, which indicated that the data could be used for subsequent analyses. DEGs of key enzymes were analyzed using the transcriptome annotation results and the main monoterpenoid biosynthetic of the KEGG metabolic

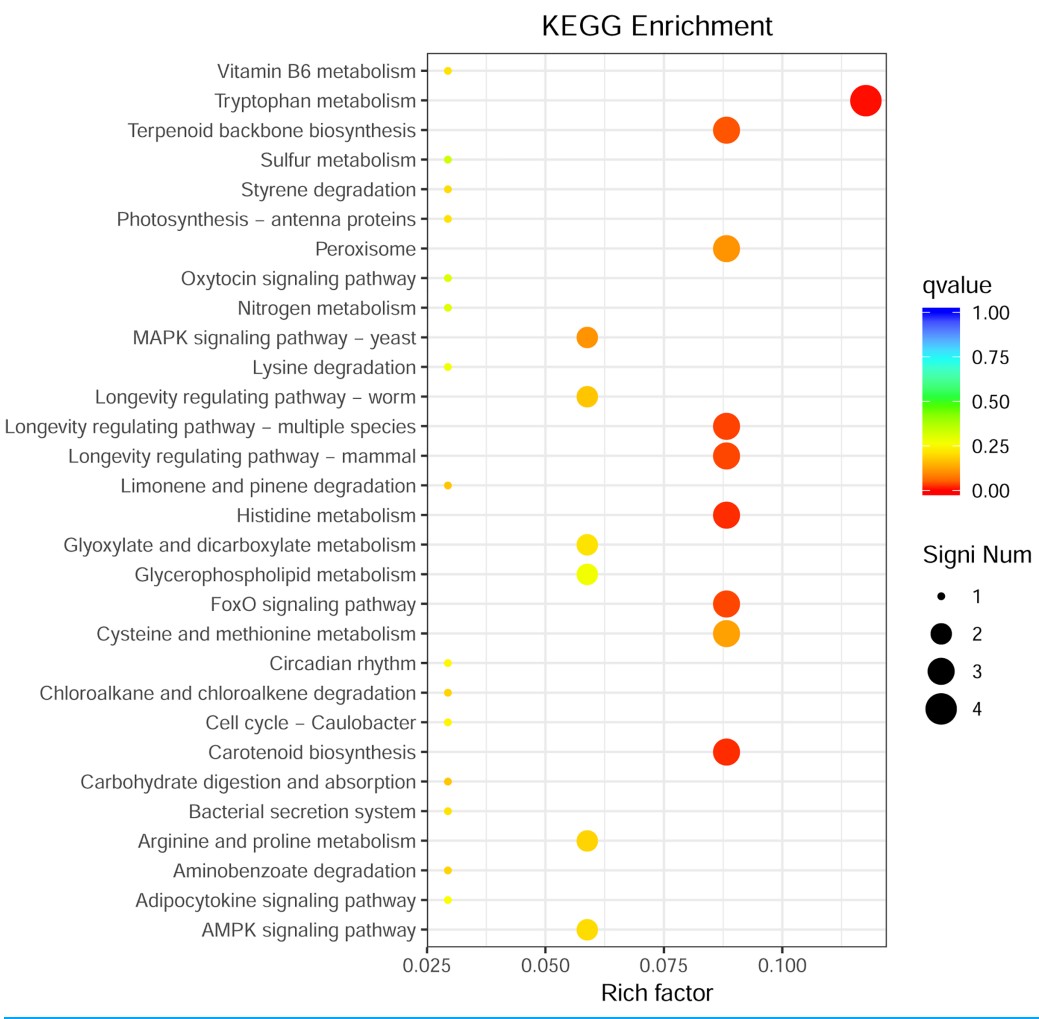

**Figure 5  KEGG metabolic pathway enrichment.**

pathway. Significant differences were found for genes *GPPS*, *LS*, *L3OH*, *iSPD*, *PR*, *MD*, *NMD*, and *L6OH*. Statistics are shown in Table S7.

All relative expression analyses of the selected unigenes were performed based on M8, the *l*-menthol chemotype. In the comparison groups of different chemotypes, we observed that the relative expression of enzyme genes in the monoterpenoid biosynthesis pathway was related to the main components of different chemotypes of essential oil (Fig. 6). In pulegone chemotypes, the expression level of *PR* (TRINITY_DN81154_c1_g1 and TRINITY_DN81915_c0_g3) was significantly lower than for other chemotypes ($P < 0.05$). However, the relative expression level of *MD* (TRINITY_DN67710_c6_g2 and TRINITY_DN89117_c1_g2) in *l*-menthol chemotypes was significantly greater than that of other chemotypes ($P < 0.05$). Highly expressed *MD* may enable the biosynthesis of monoterpenes to eventually generate *l*-menthol. *L6OH* (TRINITY_DN74923_c4_g1) expression in carvone chemotypes was greater compared to that in other chemotypes. The major component of P8 was pulegone, the major component of P5 was menthone, and the major component of M8 was (−)-menthol. Both menthone and menthol are

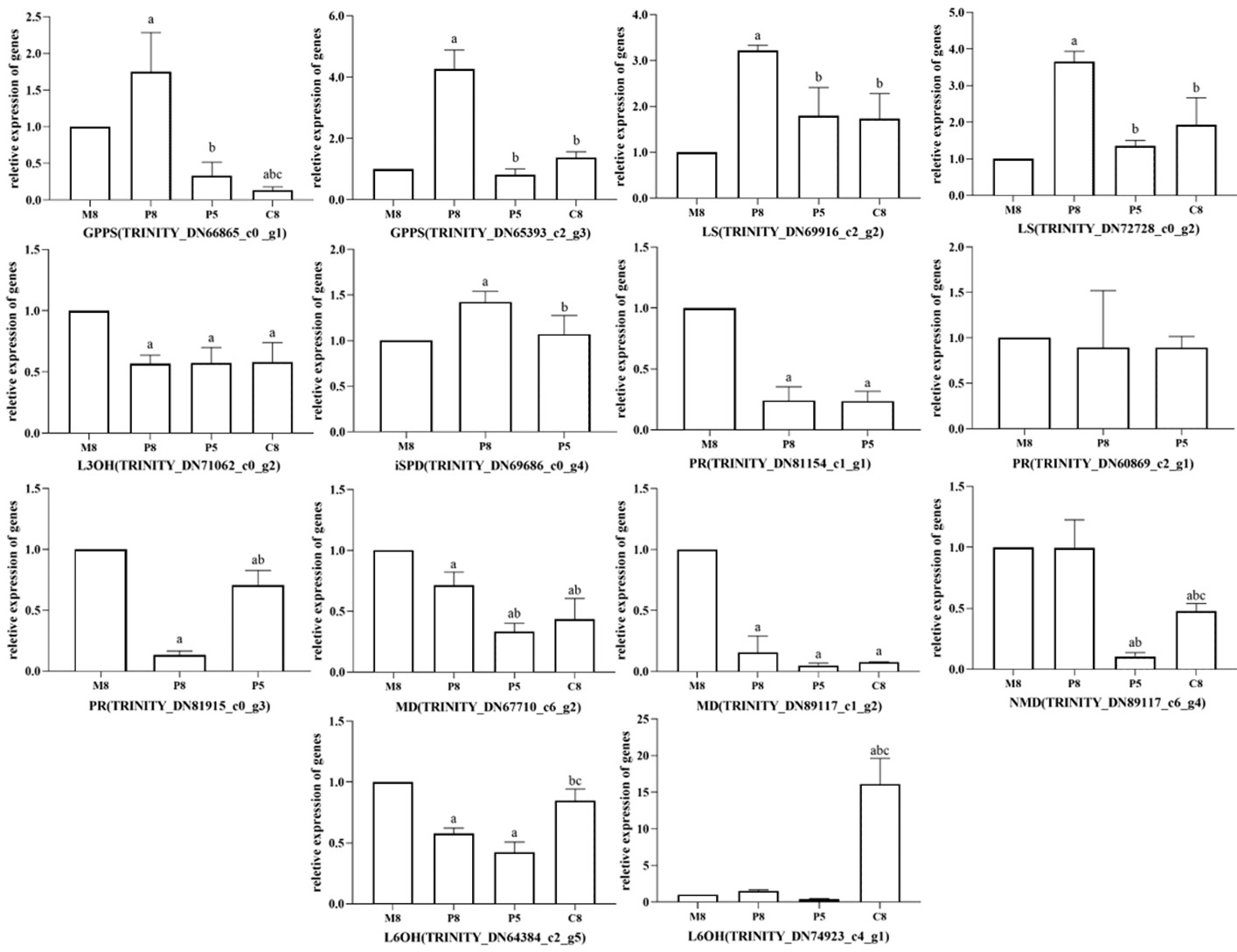

**Figure 6 Relative expression levels of key enzyme genes in different chemotypes.** The lowercase letter a indicates a statistically significant difference compared to M8 ($P < 0.05$), b indicates a statistically significant difference compared to P8 ($P < 0.05$), c indicates a statistically significant difference compared to P5 ($P < 0.05$) (Ordinary one-way ANOVA).               

downstream of PR in the monoterpenoid biosynthesis pathway. Fig. 6 shows that the relative expression of *PR* in P8 is lower than that in M8, inferring that PR maintains an important role in *M. haplocalyx*. Similarly, MD, L6OH, and L3OH are also enzymes with important positions in this pathway in *M. haplocalyx*. It could be argued that the abundance of enzyme expression might affect the variation of chemotypes.

## Identification and analysis of the differentially expressed TFs of different chemotypes

In total, 2,599 TFs from 66 families were identified in *M. haplocalyx* RNA-seq data. The C2H2, MYB, C3H, AP2/ERF, bHLH, WRKY, GRAS, MYB, bZIP, and NAC TF families were the 10 largest families (Fig. S3). The differential TFs identified in the

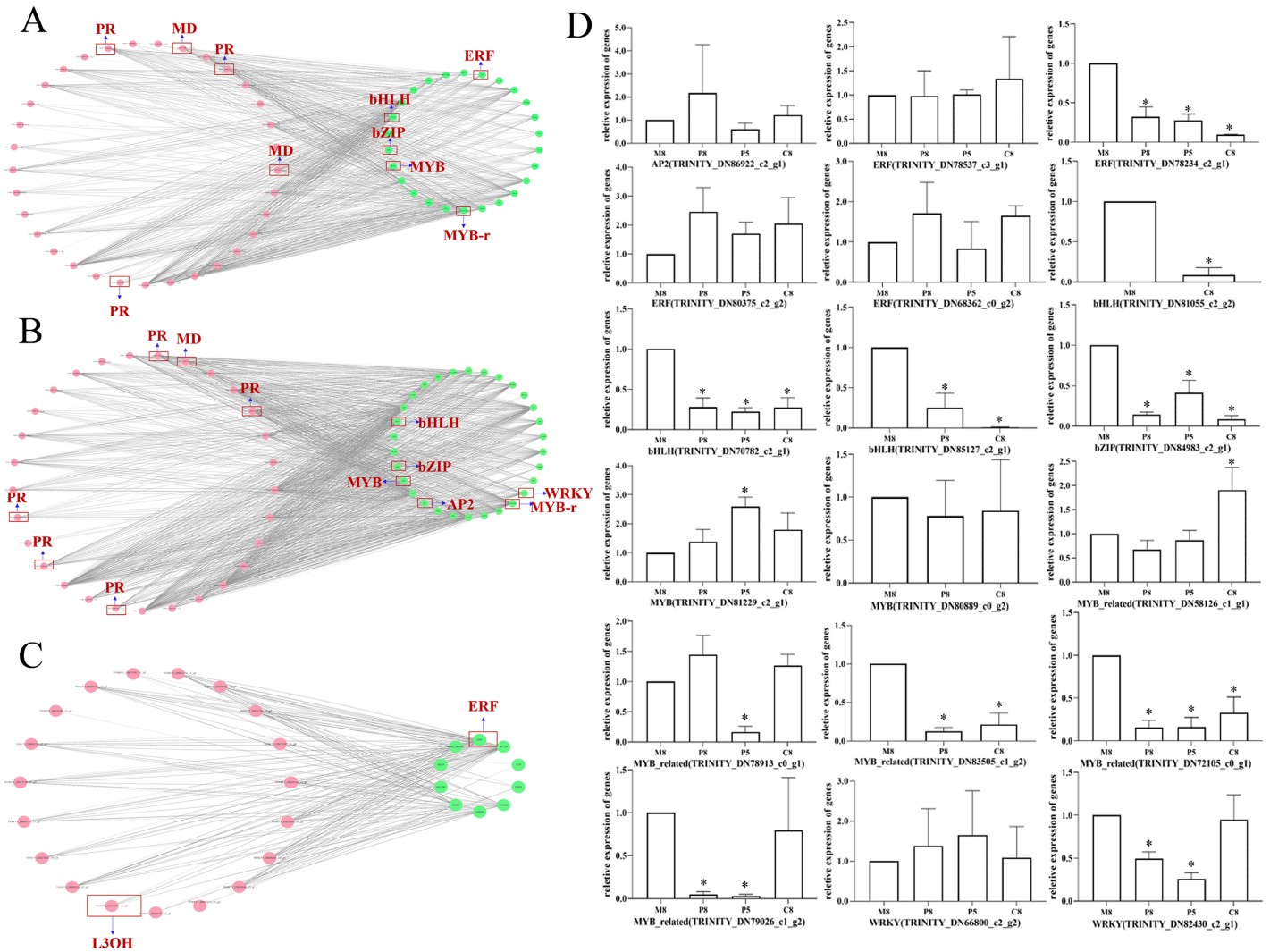

**Figure 7 The correlation network diagram between different chemotypes of key enzyme differential genes and differential TFs and relative expression levels of related TF genes in different chemotypes.** (A–C) Correlation network diagram of M8 *vs* P8, M8 *vs* P5, and M8 *vs* C8, respectively. Pink dots represent the Gene ID corresponding to the screened differential enzyme genes, green dots represent TFs, and gray lines represent that the two are correlated. (D) Relative expression levels of related TF genes in each compared group. An asterisk (*) indicates significant differences compared to M8 in relative expression levels ($P < 0.05$) (ordinary one-way ANOVA).

transcriptome data included 113 TFs from 34 families, among which GRAS, bHLH, C2H2, ERF, Dof, MYB, TCP, B3, WRKY, C3H, and Trihelix contained more than five DEGs.

## Interaction network analysis of genes related to monoterpenoid biosynthesis of different chemotypes

To further investigate the regulatory mechanism of monoterpenoid biosynthetic pathways of different chemotypes, we used Cytoscape to visualize the correlation between the key enzyme genes of monoterpenoid biosynthetic pathways and the expression profiles of TF genes (Fig. 7; Table S8). According to a past study (*Dong et al., 2020*), the main TFs involved in the regulation of terpenoid synthesis include six TF families: AP2/ERF, bHLH, MYB, NAC, bZIP, and WRKY. It is worth noting that in the comparison of M8 *vs.* P8,

*bHLH*, *bZIP*, *ERF*, and *MYB* showed very high correlations with *MD* and *PR* ($r > 0.95$) (Fig. 7A). The expression of eight unigenes of these TFs families was significantly different ($P < 0.05$, Fig. 7D). In the comparison of M8 *vs.* P5, *bHLH*, *bZIP*, *AP2*, *MYB*, and *WRKY* showed very high correlations with *MD* and *PR* ($r > 0.95$) (Fig. 7B). The expression of seven unigenes of the *bHLH*, *bZIP*, *MYB*, and *WRKY* families was significantly different ($P < 0.05$, Fig. 7D). We speculated that these TFs might affect the essential oil components in different chemotypes by regulating the expression of *MD* and *PR*. The expression patterns of these key enzymes in turn regulate the variation of *l*-menthol and pulegone chemotypes. In the comparison of M8 *vs.* C8, *ERF* showed a high correlation with *L3OH* and *L6OH* ($r > 0.85$) (Fig. 7C). *ERF* (TRINITY_DN78234_c2_g1) was significantly differentially expressed ($P < 0.05$, Fig. 7D). Therefore, these TFs may regulate the transcription of *L3OH* and *L6OH* and thus their chemotypes. The relative expression trends of related TFs in different chemotypes were verified, as shown in Fig. 7D. All relative expression analyses of the selected unigenes were performed based on M8, the *l*-menthol chemotype. Although *ERF* (TRINITY_DN78234_c2_g1), *bHLH* (TRINITY_DN70782_c2_g1 and TRINITY_DN85127_c2_g1), *bZIP* (TRINITY_DN84983_c2_g1), *MYB-related* (TRINITY_DN72105_c0_g1) were all significantly different compared to M8 ($P < 0.05$), some of them were not highly correlated with key enzymes (Table S8).

## DISCUSSION

The genus *Mentha* is rich in essential oil, which harbors the main active ingredients. Many chemotypes of essential oils have different characteristics and aromatic flavors because of the varying terpene ratios (*Salehi et al., 2018*). The chemotypes of *M. haplocalyx* have changed since most species have been transformed from cultivated to wild or from wild to cultivated, making it difficult to repeat the previous chemotype classification. Although there have been reports on the chemotypes of *Mentha* plants (*Dong, Ni & Kokot, 2015*; *Llorens-Molina et al., 2017*; *Narasimhamoorthy et al., 2015*), the mechanism of their formation has not been studied in depth. We collected different representative chemotypes of *M. haplocalyx* to be planted uniformly. From years of research on the dynamic changes of chemotypes, we found that *M. haplocalyx* chemotypes remain stable during the traditional harvest period (*Yang, 2017*). In this study, 12 sequencing libraries were constructed and assembled to obtain 254,942 unigenes, with an average length of 504 bp (Table S2). Functional annotations indicated that 47.77% of the unigenes could be annotated in at least one database. This report is the first transcriptome study of different chemotypes of *M. haplocalyx*. It will provide useful information for further study of this species.

Due to the wide application of (−)-menthol, it would be useful to select *l*-menthol chemotype-based *M. haplocalyx* for production by studying the formation of chemotypes. However, previous studies on mint metabolic engineering mostly focused on how to improve essential oil production. There are two ways to increase essential oil yield: (1) manipulating the expression levels of selected genes involved in the MEP pathway and (2) overexpressing genes that encode enzymes catalyzing seemingly slow steps in monoterpene biosynthesis. For example, *Lange et al. (2011)* generated transgenic plants,

tested under greenhouse and commercial-scale field trial conditions, in which a two-gene combination (downregulation of MFS and upregulation of DXR) positively affected oil yield and composition. We found that in *l*-menthol chemotype, the relative expression level of *MD* was significantly higher than in other chemotypes. The expression of *PR* in pulegone chemotype was lower than that of other chemotypes and in carvone chemotypes, *L6OH* expression was higher than in other chemotypes. Consistent results were found in *M. arvensis*, whose main ingredient is menthol, and *Mentha spicata*, whose main ingredient is carvone (*Ahkami et al., 2015*; *Akhtar et al., 2017*). The accumulation of main monoterpenoids of different chemotypes is associated with diverse expression patterns of key enzyme genes, which suggests that the variation of chemotypes is related to differential control or gene regulation. We speculate that the expression of different chemotype enzymes may be regulated by other factors. *Mishra et al. (2021)* came to a similar conclusion.

Transcription regulation is the core of plant secondary metabolism. The formation of monoterpene is not merely dependent on the biochemical properties of the enzymes involved in the biosynthesis but also on the contribution of the TFs. Studying how TFs regulate the transcription and expression of enzyme genes, and exploring the signaling pathways and mechanisms in which they participate, can provide a basis for in-depth research on the metabolic regulation of terpenoids. Compared with that of widely studied metabolic pathways, such as flavonoids, the transcriptional regulation of terpene metabolism has only been verified in a few studies (*Qi et al., 2018*). TFs can simultaneously participate in the expression regulation of multiple key genes in the terpenoid metabolism-related gene cluster (*Zhou et al., 2016*). The main TFs involved in the synthesis of plant terpenoids include six TF families: AP2/ERF, bHLH, MYB, NAC, bZIP, and WRKY (*Chuang et al., 2018*; *Dong et al., 2020*; *Pan et al., 2019*). *Qi et al. (2018)* speculated that, in *Mentha* plants, a large number of upregulated WRKY factors may have important functions in the transcriptional regulation of the secondary metabolism of *Mentha canadensis L*'s JA response. MsMYB can negatively regulate the biosynthesis of monoterpenes (*Reddy et al., 2017*; *Wang et al., 2016*). In this study, we found that the bHLH, bZIP, AP2/ERF, and MYB families were highly correlated with key enzymes of monoterpenoid synthesis in different chemotypes. Therefore, these TFs may have important functions in the variation of chemotypes and could thus be used in metabolic engineering as an effective strategy for chemotypes required for breeding. However, the temporal and spatial expression characteristics, functional identification, gene cloning, and other aspects of these candidate TFs require further study.

## CONCLUSIONS

The aim of this study was to explore the formation mechanism of chemotypes to provide a basis for cultivating more widely used chemotypes of *M. haplocalyx*. We dynamically monitored the characteristic components of the essential oils of different chemotypes during different growth periods and found that *l*-menthol, pulegone, and carvone chemotypes were stable during the traditional harvest period. We selected stable chemotypes for transcriptome sequencing by analyzing the expression levels of enzyme

genes in the monoterpenoid biosynthesis pathway as the key enzyme genes expression pattern of this pathway is related to the main essential oil components in different chemotypes. The TF families of bHLH, bZIP, AP2/ERF, MYB, and WRKY were highly correlated with key enzymes of monoterpenoid synthesis in different chemotypes ($r > 0.85$). It is reasonable to infer that these TFs may play an important role in the variation of *M. haplocalyx* chemotypes. Our findings provide a list of candidate TFs for further metabolic engineering research and have reference value for revealing the formation mechanism of *M. haplocalyx* chemotypes.

### Funding

This work was supported by the Key project at central government level: The ability establishment of sustainable use for valuable Chinese medicine resources (2060302), and The National Key R&D Program of China "Chinese-Australian" 'Belt and Road' Joint Laboratory on Traditional Chinese Medicine for the Prevention and Treatment of Severe Infectious Diseases (Grant Number: 2020YFE0205100). The funders had no role in study design, data collection and analysis, decision to publish, or preparation of the manuscript.

### Grant Disclosures

The following grant information was disclosed by the authors:
Key project at central government level: 2060302.
National Key R&D Program of China "Chinese-Australian" 'Belt and Road' Joint Laboratory on Traditional Chinese Medicine: 2020YFE0205100.

### Competing Interests

The authors declare that they have no competing interests.

### Author Contributions

- Xin An conceived and designed the experiments, performed the experiments, analyzed the data, prepared figures and/or tables, authored or reviewed drafts of the article, and approved the final draft.
- Jingqiong Wan performed the experiments, analyzed the data, prepared figures and/or tables, authored or reviewed drafts of the article, and approved the final draft.
- Hui Jiang performed the experiments, analyzed the data, authored or reviewed drafts of the article, and approved the final draft.
- Yangzhen Liao analyzed the data, authored or reviewed drafts of the article, and approved the final draft.
- Chang Liu analyzed the data, authored or reviewed drafts of the article, and approved the final draft.
- Yuan Wei conceived and designed the experiments, authored or reviewed drafts of the article, and approved the final draft.
- Chongwei Wen analyzed the data, authored or reviewed drafts of the article, and approved the final draft.

- Zhen Ouyang conceived and designed the experiments, authored or reviewed drafts of the article, and approved the final draft.

## DNA Deposition
The following information was supplied regarding the deposition of DNA sequences:
All sequences are available at NCBI: PRJNA795820.

## Data Availability
The raw measurements are available in the Supplemental Files.

## Supplemental Information
Supplemental information for this article can be found online at http://dx.doi.org/10.7717/peerj.14914#supplemental-information.

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
