# Peer review of "Transcriptome analysis of transcription factors and enzymes involved in monoterpenoid biosynthesis in different chemotypes of Mentha haplocalyx Briq"

_PeerJ, doi:10.7717/peerj.14914_

## Round 0.1 · original submission · Major Revisions

The reviewer comments have now been received. As you see, they have given several suggestions to improve the work. Kindly go through and address them point by point.

Details of methodology, primer details, kits used for cDNA synthesis etc need to be mentioned. Additionally, I would request alteration in the title (use either enzymes or genes) and English editing where appropriate.

Pooling of the reads and downstream analysis is not satisfactory and the reviewer is very skeptical about it.

Without additional experiments like yeast two hybridization assay, results are not conclusive and manuscript acceptance will require validation of results, clear expression along with detailed methodology.

Best of luck with your endeavours.

Reviewer 1 ·

Basic reporting

Overall, this paper is clear and well-written. The introduction of background is thorough and the literature is sufficiently cited.

Experimental design

In this work, the authors sought to utilize transcriptome sequencing to identify key transcripts responsible for different chemotypes of Mentha haplocalyx. It will be interesting for the broad readership at PeerJ. The research question is clearly articulated, and the experiments are well-designed. I only have a few comments on data analyses and presentations:
1. It is unclear to me how Figure 4 and Figure 5 GO terms and KEGG analyses were done, with which set of DEGs?
2. The figure 4 y-axis and the messages behind the bar plots are confusing, the authors may want to do similar plots like figure 5 of KEGG analyses.
3. In figure 7abc, how the interaction network was established is not clear enough to me. The authors shall make it clear what each node and line stands for in the figure.

Validity of the findings

In figure 7D panel:
1. The error bar of the black bar is missing (M8).
2. The statistical testing here between each comparison is also missing.
3. Please explain why some genes only have three bars or two bars while others have four bars.

Reviewer 2 ·

Basic reporting

The authors have shown the importance of the TFs which might be playing an interactive role in the chemotype variation in M. haplocalyx. The manuscript is clear to some extent and the English language can be still improved. The introduction and background is clear, references are valid. The manuscript is basically a preliminary finding which can be scientifically relevant for futuristic aspects of the study. However, the authors are requested to follow the comments and suggestions for the overall improvement of the manuscript.

The Background information can be improved. It should be precise and informative. In the Methods section defining the description of “chemotype formation” can be alternatively written as “To further investigate the chemotype variation, …”. Similarly the phrase can be changed at other sections in the manuscript.

Experimental design

Methods :
Line 118: The authors can explain the details of the type of samples collected, growth stage and the growth conditions in a more elaborative manner.
Line 130: The authors have mentioned using different chemotypes sampled at different time however, Line 118 do not mention the same. Thereby the details should be included for transcriptome and oil content analysis separately and clearly.
Line 133: Kindly mention the RIN value and the sample type used to make the library.
Line 162: The authors should mention the method used for relative quantification of the gene expression.
Results:
Line 172: The authors should mention the early growth/ vegetative stage used in the analysis and why they have narrowed preferentially on these time period (Figure 2)
Line 180-182: The authors can explain as to why they have chosen the particular months for transcriptome in comparison with the Figure 2 to provide a clear intention to the readers as the age and the time point for transcriptome analysis is varying for the chemotypes.
Line 221: The authors should clearly indicate the relative expression analysis of the selected genes and the relative reference chemotype, which is l-menthol chemotype-based M. haplocalyx. The figure 6 does not clearly point to the statistical changes as compared to the reference chemotype. The authors can improve the figure 6 for a better understanding. For example: The different letters for the comparative chemotype is not given and makes the statistical difference comparison not so clear.
Line 254 and 255: Same goes for the relative analysis. Kindly include the statistical significance among the four parameters.

Discussion
The discussion section can be improved.

Validity of the findings

The authors initially focus on the chemotype variation with reference to the transcriptomic changes observed related to the TFs, however, the explanation and correlation of the findings can be made little elaborative as to how they have contributed to the scientific repository wrt their transcriptome analysis and expressional analysis. The authors have not shown any experimental correlation of these TFs contributing to the chemotypic nature thus they can cite references related to the experimental works focusing on the candidate genes in related or model species for justifying the relatedness of their findings.

·

Basic reporting

Except for a few sentences and wordings that should be revised for better clarity, the manuscript is clear and concise. Below are some are examples:

Line 20: “The main active ingredients of Mentha haplocalyx Briq. are essential oils.”, do you mean monoterpenes? Essential oils are composed of ingredients likes terpenes.
Line 47-48: “Chemotype is a form of biodiversity within plant species (Guo et al., 48 2008; Hua et al., 2009).” The definition isn’t enough. Consider rephrasing as follows or something similar - “Chemotype is a form of biodiversity within essential oil producing plant species like Mentha based on their chemical profiles (Guo et al., 48 2008; Hua et al., 2009).”
Line 95: The transition from describing the different chemotypes to TFs implies that these are the only factors regulating the chemotype of the cultivar. Introduce other types of mechanisms that could play a role and describe your hypothesis as to why TFs were selected in this study.
Line 94-95: “Transcription factors (TFs) regulate the coordinated expression of enzyme genes in the synthesis pathways of secondary metabolism.” Consider rephrasing as follows or something similar - “Transcription factors (TFs) regulate the coordinated expression of genes involved in the biosynthetic pathways of secondary metabolism.”
Line 108-110: “The aim of this study was to explore the formation mechanism of chemotypes to provide a basis for cultivating more widely used chemotypes of M. haplocalyx.” Consider rephrasing as follows or something similar - “The aim of this study was to explore the role of TFs in determining different chemotypes to provide a basis for cultivating more widely used chemotypes of M. haplocalyx.”

Experimental design

In general the methodology needs more description. There are many referenced methods without brief description and that needs to be addressed. One area of concern is the post assembly processing. Although not clear from the way it is written, results imply that reads from all experiments were pooled and assembled. Furthermore, alternative splice variants were not removed. There is no problem with that but this complicates the interpretation of results which will be covered next. here few more comments:

Line 124-127: briefly describe both extraction and GS-MS methods used
Line 129-140: Your methodology should answer which library prep did you use? which illumina platform was used for sequencing? Whether paired or single end sequencing was used? What was the fragment length? etc … Also what do you mean by “Random primers were used to reverse transcribe fragments of mRNA into cDNA and the second strand cDNA was then synthesized. Finally, end repair was performed and poly(A) was added to prepare the library.”? Can’t say much because don’t have the prep kit info here but your description of cDNA synthesis might need to be corrected.
Line 142: what were the Trimmomatic parameters? What was the size of the sequencing output and what was the size of your library after cleaning? Did you pool all samples for assembly? Did you filter duplicates and how did you handle splice variants, i.e. does the number of unigens output refers to the longest read or all unigenes? This is important, if in case splice variants play a role in enzyme activity
Line 148-151: the sentence isn’t clear
Line 157: what was your comparator for fold change calculation? Found in the result that you used M8 as a comparator chemotype, any particular reason for that or was this selected randomly?

Validity of the findings

Figure six is the central piece of the manuscript and the most problematic in my eyes because instead of addressing the objective of the experiment it showed the possible role for other regulation mechanism, i.e. splice variants, in determining chemotypes. For example, from the three PR transcripts only one (........c1_g1) was clearly downregulated, the other two are either not consistent between the two P chemotypes or not significantly downregulated. This makes it difficult to a) exclude the role of splice variants in chemotype regulation and since the reads were pooled we can not conclusively show which splice variant was associated with which chemotype, b) to associate the TFs to specific transcripts. Similarly, two MDs were detected but only one displayed significant and consistent downregulation (MD(Trinity.....V1_g2). Also, the level of downregulation of both PR and MD genes (< two fold) doesn't reflect the essential oil profile/chemotype. L6OH also showed similar pattern but at least c4_g1 upregulation is the only one that is clearly downregulated and to the expected level for RNAseq and qPCR level detection. This makes the interpretation of the in silico association results questionable.

I also expected additional results, example yeast two hybrid or pull down assays or any other biological assay result, to validate the in silico association result before making a concrete conclusion. Therefore for the in silico result the authors generated, the TF/chemotype association conclusion is a stretch and not justified.

Additional comments

Few more additional points:

First the authors seem to mix up methodology and results. One such example is the sequencing platform was given in the result section not in methods, comparator for fold expression calculation was never mentioned in the method section etc.

Some results are missing, like qPCR validation

I strongly encourage the authors to introduce other possible regulatory mechanism that could determine chemotype of a cultivar. I didn't read the discussion section because, the revised results will impact the discussion significantly.

---

## Round 0.2 · accepted · Accept

The reviewer is now satisfied by the response of authors and the manuscript may now be published.

Reviewer 2 ·

Basic reporting

The authors have updated the suggested comments

Experimental design

The authors have included the suggestions in the updated version.

Validity of the findings

The validity needs the confirmatory experimental evidences aimed for future work by the authors. The current findings include preliminary findings.

Additional comments

NO comments